# Cannabidiol-Loaded Nanocarriers and Their Therapeutic Applications

**DOI:** 10.3390/ph16040487

**Published:** 2023-03-24

**Authors:** Elham Assadpour, Atefe Rezaei, Sabya Sachi Das, Balaga Venkata Krishna Rao, Sandeep Kumar Singh, Mohammad Saeed Kharazmi, Niraj Kumar Jha, Saurabh Kumar Jha, Miguel A. Prieto, Seid Mahdi Jafari

**Affiliations:** 1Food Industry Research Co., Gorgan 49138-15739, Iran; 2Food and Bio-Nanotech International Research Center (Fabiano), Gorgan University of Agricultural Sciences and Natural Resources, Gorgan 49138-15739, Iran; 3Department of Food Science and Technology, School of Nutrition and Food Science, Food Security Research Center, Isfahan University of Medical Sciences, Isfahan 81746-73461, Iran; 4Department of Pharmaceutical Sciences and Technology, Birla Institute of Technology, Mesra, Ranchi 835215, India; 5School of Pharmaceutical and Population Health Informatics, DIT University, Dehradun 248009, India; 6Faculty of Medicine, University of California, Riverside, CA 92679, USA; 7Department of Biotechnology, School of Engineering and Technology (SET), Sharda University, Greater Noida 201310, India; 8School of Bioengineering and Biosciences, Lovely Professional University, Phagwara 144411, India; 9Department of Biotechnology Engineering and Food Technology, Chandigarh University, Mohali 140413, India; 10Department of Biotechnology, School of Applied and Life Sciences (SALS), Uttaranchal University, Dehradun 248007, India; 11Nutrition and Bromatology Group, Department of Analytical Chemistry and Food Science, Faculty of Science, Universidade de Vigo, E-32004 Ourense, Spain; 12Department of Food Materials and Process Design Engineering, Gorgan University of Agricultural Sciences and Natural Resources, Gorgan 49189-43464, Iran; 13College of Food Science and Technology, Hebei Agricultural University, Baoding 071001, China

**Keywords:** *Cannabis sativa*, cannabidiol, molecular mechanism, bioavailability, nanomedicine

## Abstract

Cannabidiol (CBD), one of the most promising constituents isolated from *Cannabis sativa*, exhibits diverse pharmacological actions. However, the applications of CBD are restricted mainly due to its poor oral bioavailability. Therefore, researchers are focusing on the development of novel strategies for the effective delivery of CBD with improved oral bioavailability. In this context, researchers have designed nanocarriers to overcome limitations associated with CBD. The CBD-loaded nanocarriers assist in improving the therapeutic efficacy, targetability, and controlled biodistribution of CBD with negligible toxicity for treating various disease conditions. In this review, we have summarized and discussed various molecular targets, targeting mechanisms and types of nanocarrier-based delivery systems associated with CBD for the effective management of various disease conditions. This strategic information will help researchers in the establishment of novel nanotechnology interventions for targeting CBD.

## 1. Introduction

The plant cannabis or *Cannabis sativa* L. is among the plants with the longest histories of cultivation for its medicinal properties. It produces many phytochemicals such as cannabinoids, terpenes, and flavonoids, and is a major source of cannabinoids including cannabidiol (CBD), cannabinol (CBN), cannabigerol (CBG), tetrahydrocannabinol (THC), and cannabichromene (CBC) [1]. Initial use of cannabis dates back 5000 years in China [2]. Cannabinoids exhibit numerous potential pharmacological activities against nausea, epilepsy, inflammation, depression, and other clinically relevant effects [3]. Recently, there has been an increasing global acceptance of cannabinoids, due to the decoding of its genome description and molecular genetics. Moreover, the determination of the possible heteromeric complex formations, along with crystal structure determination of the CB1R and CB2R receptors, has led to the discovery and subsequent understanding of the human endocannabinoid system (ECS). The ECS refers to the cannabinoid receptors CB1, CB2, CB3, and other associated compounds that are activated by cannabinoids derivatives, which may include cannabinoids, endocannabinoids, other related compounds, along with their metabolic enzymes. Furthermore, detailed study involving cellular, molecular, biochemical and behavioral responses have revealed that CNR1 and CNR2 genes are mapped by the human chromosome 6 and 1, respectively, and are encoded by the use of cannabis and cannabinoids [4]. 

CBD usually contains a tetrahydro-biphenyl skeleton. A bicyclic core represents an adduct formed by the monoterpene, *p*-cymene, and the alkylresorcinol derivative olivetol [5]. Moreover, via an acid-catalyzed reaction, it can be converted into the tricyclic dibenzopyrane, Δ9-THC (THC) [6]. Although CBD and THC are predominantly found in plants of the cannabis genus, they just form a part of the bigger terpenyl-alkylresorcinol chemical subspace. Cannabinoids’ natural expression undergoes tremendous changes during its growth and usually consists of four major classes of molecules, each characterized by distinct skeletons [6]. All classes contain an olivetol core in which a hydroxyl group is replaced by an alkyl chain (Figure 1). 

In addition to the bicyclic tetrahydro-diphenyl and tricyclic dibenzopyrane classes, benzopyranes (prototype: cannabichromene [CBC]) and acyclic prenyl-olivetols (prototype: cannabigerol [CBG]) are the other key classes of cannabinoids. These can undergo further structural modifications, expanding into five major dimensions: (i) variation of terpenyl saturation and linkage; (ii) terpenyl stereochemistry; (iii) presence vs. absence of C-2 carboxylation (“acids”); (iv) variation of length and branching of the olivetol alkyl chain (e.g., homologous cannabinoids) [7]; and (v) presence and variation of the monoterpene vs. sesquiterpene moieties. These collectively give some natural cannabinoid metabolites, as they are only one family among several other phytochemical classes in cannabis plants.

## 2. Cannabidiol (CBD): An Overview

Cannabidiol (CBD) is therapeutically used in a number of diseases such as Alzheimer’s disease/dementia, glaucoma, traumatic brain injury/spinal cord injury, chemotherapy-induced nausea/vomiting, appetite and weight loss, chronic pain, cancer, Huntington’s disease, Parkinson’s disease, dystonia, addiction, anxiety, depression, irritable bowel syndrome, epilepsy, spasticity of multiple sclerosis, Tourette syndrome, amyotrophic lateral sclerosis, sleep disorders, posttraumatic stress disorder, and schizophrenia. CBD, encapsulated in either nanocarriers or as advanced delivery systems, are frequently employed to overcome limitations faced by traditional forms. A few CBD formulations are illustrated in Table 1.

### 2.1. Bioavailability and Safety of CBD and Associated Derivatives

The bioavailability of CBD mainly depends upon its mode and route of administration [24,25]. CBD is mostly formulated as a solution, either in oil or alcohol form, which is then processed as either soft gelatin capsules, oral solution, oromucosal spray or sublingual drops. There is huge uncertainty concerning the bioavailability of CBD administered through different routes; CBD and THC administered via oral and oromucosal routes exhibit high inter/intra-individual variability. Overall, the oral bioavailability of CBD is estimated to be around 6–9%, with *t_max_* hovering between 1–4 h. A gelatin capsule containing 20 mg CBD exhibited a half-life = 1.4–10.9 h, and *C_max_* = 2.4 ng/mL. Administration of CBD along with a high calorie/fat meal is shown to increase its oral bioavailability and is one of the potential techniques used to increase its bioavailability. This has been tested in several individuals, and subjects in fed state exhibited 4- and 14-fold increase in bioavailability and *C_max_*, respectively [26,27]. 

### 2.2. Extraction

Extraction in the hemp industry is mainly performed in two ways: (a) extraction from trichomes, and (b) extraction from seeds. Extraction from trichomes is mainly done to obtain medicinal or recreational products. It mostly yields cannabinoids and terpenes. By contrast, in seed extraction, fatty acids are extracted, which finds use as biodiesel and in the cosmetics and food industries. Cannabinoids and terpenes are extracted from female flower buds just before pollination using polar solvents, as the concentration of cannabinoids are at peak levels just before flowering. The most employed methods for extraction include organic solvent extraction (OSE), supercritical fluid extraction (SFE), and Soxhlet apparatus. Mild comminution can be done for a better yield. Care should be taken to avoid excessive grinding as it may lead to the co-extraction of undesirable products [28,29]. Usually, along with the main components THC and CBD, waxes and pigments are also co-extracted.

A number of studies employing Soxhlet apparatus have extracted cannabinoids [1]. In the OSE method, the plant biomass is left immersed in an organic solvent for a defined period, for an effective mass transfer, whereas the SFE method uses the safe and capable solvents in their critical state to efficiently extract CBD. SFE enables selective extractions by increasing the temperature and pressure to critical state, where no liquid can be further liquefied at higher pressure, further increasing the liquid density [14]. Various solvents such as ethene, water, methanol, carbon dioxide, nitrous oxide, sulfur hexafluoride, *n*-butene, and *n*-pentane are commonly employed in SFE [14]. 

### 2.3. Biological Effects of CBD

#### 2.3.1. Roles of CBD in the Immune System

CBD has been found to interact with the immune system in several ways, both positive and negative. The immune system is responsible for protecting the body from harmful pathogens and other foreign invaders. One of the most promising areas of research on CBD and the immune system is its potential to reduce inflammation. Inflammation is a natural response of the immune system to injury or infection, but chronic inflammation can lead to a host of diseases, including cancer, heart disease, and autoimmune disorders [30,31].

CBD has been found to have anti-inflammatory properties, mostly proved in several in vitro and in vivo disease models of inflammation. CBD has been proved to interact in inflammation processes through different mechanisms, both by innate and adaptive responses: inhibiting the production of inflammatory cytokines (IL-1α, IL-1β, IL-6, TNF-α and IL-17A); reducing the production of nitric oxide (NO) or myeloperoxidase (MPO) pro- from innate cells; involving suppression of inducible nitric oxide synthase (iNOS); mediating apoptosis in lymphocytes contributing to immune suppressive mechanism; and decreasing microglial cell activation, among others [30,31]. On the other hand, it is important to highlight the relationship between inflammation and redox imbalance. In this regard, CBD has been shown to affect this balance by modifying the enzymatic activity of anti- and pro-oxidants, as well as the occurrence of other processes such as transition metal ions chelation, micronutrients supporting antioxidant activity, or oxidative modifications of lipids, proteins, or DNA [4].

CBD has also been studied in the context of autoimmune disorders. Some research suggests that CBD may be able to modulate the immune response in these conditions, potentially reducing symptoms and improving quality of life. The role of CBD in autoimmune diseases acts through microglial activation, T-cell proliferation, and suppressing the actions of proapoptotic proteins [30,31] However, it is important to note that the research on CBD and the immune system is still in its early stages, and more research is needed to fully understand its effects on physiological immune response.

#### 2.3.2. Roles of CBD in the Nervous System

Cannabidiol (CBD) has gained attention in recent years for its potential therapeutic benefits, particularly in the nervous system. CBD interacts with the body’s endocannabinoid system (ECS), which plays a crucial role in regulating various physiological processes including sleep, appetite, pain, immune function and nervous system derived pathologies. CBD interacts with the ECS by binding to non-psychoactive receptors such as CB1 and CB2 [31] Pre-clinical research shows that CBD has ability to reduce anxiety or depression and improve mood, not acting through the CB1 receptor but interacting with other targets involved in neurologic disorders. However, the therapeutic effectiveness of CBD in anxiety disorders needs to be claimed in further clinical trials [31]. 

CBD also exhibits analgesic and neuroprotective effects, protecting neurons from damage and death caused by inflammation and oxidative stress. This could be beneficial in the treatment of neurological disorders such as Alzheimer’s and Parkinson’s disease or multiple sclerosis [31]. The mechanisms responsible for the potential analgesic and neuroprotective effects of CBD in nervous system disorders are not entirely understood [32]. For example, in the case of analgesic effects, it is known that the anxiolytic properties of CBD may have an influence on the occurrence of pain. The mechanisms that CBD uses to protect neurons are diverse. Some of them are related to the ECS (CB1 and CB2) system but many others use non-CBD-mediated mechanisms: serotonin receptors by 5HT1A facilitation, oxidative stress, mitochondrial dysfunction, peroxisome proliferator-activated receptor gamma (PPAR-γ), inflammatory changes (modulating pro-inflammatory cytokines and elevation of brain-derived neurotrophic factor [BDNF] levels), upregulation of the mRNA levels for Cu–Zn superoxide dismutase (SOD), preventing apoptotic signaling via a restoration of Ca^2+^ homeostasis, and β-amyloid (Aβ) aggregation and stimulation by upregulating the Wnt/b-catenin pathway [31,32,33].

CBD may also play a role in the treatment of epilepsy. Studies have shown that CBD can reduce the frequency and severity of seizures in individuals with epilepsy. This led to the FDA’s approval of the first CBD-based medication, Epidiolex, for the treatment of seizures associated with certain rare forms of epilepsy [34]. It is hypothesized that these beneficial properties could be linked to a “multi-target drug” profile involving some of the mechanisms explained before and more. Nevertheless, clinical trials are still ongoing to support this evidence [35].

#### 2.3.3. Roles of CBD in the Cardiovascular System

CBD has been studied for its potential medicinal properties, including its beneficial effects on the cardiovascular system. CBD can be involved in various cardiovascular diseases (CVD) including myocardial ischaemia/reperfusion injury, heart failure, atherosclerosis, hypertension, and arrhythmias [31,36]. 

As previously stated, CBD has been shown to have anti-inflammatory and antioxidant effects, which may help to protect the cardiovascular system. Studies in animals and cell cultures have found that CBD can generally reduce the formation of blood clots, lower blood pressure, and improve blood flow, and more specifically, enhance vasorelaxant responses, reduce the vascular hyperpermeability, and protect against ischaemia-reperfusion damage and cardiomyopathy associated with diabetes. Additionally, CBD has been shown to reduce the risk of heart attack and stroke by reducing the activity of certain enzymes that contribute to the formation of plaques in the arteries and influencing the survival, death, and migration of white blood cells [37].

These effects are usually due to CBD’s direct interactions with the ECS (CB receptors) through agonistic and antagonistic action, by promoting or inhibiting the release of neurotransmitters, respectively [31]. However, CBD can also act indirectly by modulating the concentration of active compounds, i.e., inhibiting adenosine, thymidine, glutamate, serotonin, dopamine, and noradrenaline uptake [38]. Additionally, CBD has been also proposed to have a vascular effect through the release of vasorelaxant mediators as nitric oxide (NO) and/or actions mediated by vascular enzymes as well as cycloxygenase isoenzymes (COX) and superoxide dismutase (SOD) [31,38].

Some studies have found that CBD can lower the risk of atherosclerosis and have a beneficial effect on the cardiovascular system by reducing the risk of obesity-related CVD. While more research is needed to fully understand the mechanisms by which CBD may benefit the cardiovascular system, it should be noted that, although this compound has potential, it is not a substitute for traditional medical treatment.

### 2.4. Therapeutic Potential of CBD and Its Derivatives in Various Diseases and Disorders

The potential therapeutic activity was categorized into several stages, based on the stage of evidence such as conclusive evidence, substantial evidence, moderate evidence, limited evidence, and insufficient evidence. CBD is used therapeutically for several diseases, such as epilepsy, seizures, depression, anxiety, cancer, schizophrenia, spasticity, and chronic pain; some of them are discussed briefly in this section.

#### 2.4.1. Addition Disorder

A study conducted over a 10-week period reported that psychological symptoms and cognitive skills were improved following administration of 200 mg cannabis [39]. Similar results were also observed in regular cannabis consumers, suggesting that CBD may be useful as an adjunct treatment in cannabis dependence. Chronic use of cannabis has been shown to restore structural brain damage, as they increase the volumes of subicular and CA1 subfields [40]. Moreover, CBD chronic use is also subjected to low abuse liability, further effective in treating cannabis addiction [41]. Even though the exact mechanisms by which CBD exerts its therapeutic action is not yet elucidated, it is known that it interacts with a wide range of receptors, enzymes, and other targets, highlighting the agonist pathway of 5-HT_1A_, TRPV1, PPARγ and CB receptors, or the antagonistic role of GPR55 receptors, among others [42].

#### 2.4.2. Epilepsy

The exploration of CBD’s anti-epileptic properties dates back to the 1980s. CBD’s activity was evaluated in a human subject with uncontrolled epilepsy. When tested in human trials, 88% of subjects exhibited fewer convulsions post-CBD administration at a rate of 200–300 mg/day, while only 38% exhibited a partial cure [43]. Data from further studies revealed that CBD, when used in conjugation with other anti-epileptic drugs, decreased the seizure frequency in people with intractable seizures and Dravet’s and Lennox-Gastaut syndrome [44]. Although different mechanisms of action have been proposed, the exact mechanism underlying medically refractory disease in a high proportion of patients is still unknown [31,45].

#### 2.4.3. Anxiolytic

CBD has exhibited promising results in treating anxiety. They mainly act by activating limbic and paralimbic regions of the brain. Multiple works including in vitro and in vivo studies have concluded similarly, that CBD has therapeutic effects on depression, anxiety, and other associated psychotic symptoms. Moreover, further studies have revealed that CBD acts as a therapeutic agent only at lower doses, while at higher doses its anxiolytic activity is absent [46]. In vivo studies showed that CBD at a dose of 3 mg/kg in a mice model exhibited anxiolytic effect, whereas a CBD dose between 3–10 mg/kg exhibited cell proliferation and neurogenesis, exerting antidepressant effects. By contrast, CBD pharmacological activity was absent at higher dose of 10–30 mg/kg in vivo [47]. Overall, it was noticed that CBD activity on anxiety exhibits a U-shaped dose response curve.

#### 2.4.4. Cancer

Exploration of the anti-cancer activity of CBD has gained widespread attention among researchers. CBD in murine colorectal cell lines has exhibited various chemo preventive mechanisms. These include increasing endocannabinoid concentration, protecting DNA from oxidative damage, and reducing cell proliferation in TRPV1-, CB1-, and PPARγ-antagonists [48]. Similarly, CBD induced apoptosis, inhibited cell viability, and elevated reactive oxygen species (ROS) levels in human prostate carcinoma when administered at a dose of 1–10 µm in vivo. CBD’s role in inducing cell death and enhancing glioblastoma radiosensitivity was reported in another study. Glioblastoma effected cells when treated with CBD, induced apoptosis by upregulating these arrested cells. Moreover, it also blocked cell proliferation and produced pro-inflammatory cytokines, improving the effectiveness of CBD [6,49]. A recent systematic review highlighted the conflicting results due to different concentration exposure, administrations and time points, and stated three main targets: CB1 receptor related to viability effects, CB2 to apoptosis, and TRPV1 to inflammation and invasiveness [50].

#### 2.4.5. Chronic Pain

Cannabis or cannabinoids administration have been reported to relieve associated symptoms of pain in patients suffering with chronic pain. Although it is believed that CBD works as an analgesic by agonistically binding with supraspinal receptors CB1 and CB2, there is a lack of conclusive literature to prove the same [51]. Moreover, recent in vitro clinical studies have reported that CBD has direct agonistic activity on numerous cell surface receptors such as the adenosine A2A receptor, serotonin 1A receptor (5-HT_1A_), and the peroxisome proliferator-activated gamma (PPAR-γ) receptor, which acts as an analgesic pathway of CBD [22,52].

#### 2.4.6. Neuroprotection

The antioxidant and anti-inflammatory properties exhibited by CBD open a new paradigm for neuroprotection and subsequent reduction in hippocampal volume [39,40]. Chronic use of THC offers protection against hippocampal pathology [53]. The restorative effect offered by chronic use of CBD on hippocampus structures drives new perspectives in treatment of other neurological disorders like schizophrenia and Alzheimer’s [40]. In vivo studies on human subjects have elucidated the therapeutic effects of CBD on schizophrenia [54,55], parkinsonism [56], and other major depressive disorders [40]. Similarly, animal studies on Alzheimer’s have shown relief in symptoms, following the administration of CBD [57]. 

#### 2.4.7. Spasticity

Spasticity is one of the multiple conditions common to chronic neuroinflammatory diseases, such as multiple sclerosis. Many controlled studies, backed up with placebos, have been conducted to evaluate the effects of CBD on spasticity. Extracts of cannabis or Sativex^®^ (1:1 THC: CBD extract), when administered at a dose of 2.5–120 mg/daily, relieved spasticity symptoms in effected patients. Those exhibiting some troublesome symptoms were reduced following administration of CBD, as measured by Visual Analogue Scale (VAS) [58].

#### 2.4.8. Anti-Psychotic

CBD is extensively studied for its antipsychotic effects on schizophrenia [55]. In a study, it was shown to inhibit endocannabinoid anandamide degradation. Further, to evaluate its antipsychotic effects when compared with a potent anti-psychotic amisulpride, subjects undergoing CBD treatment exhibited an increase in serum anandamide concentrations [54]. Similarly, another study evaluating the effectiveness of a 1000 mg/daily dose of CBD showed a decrease in positive psychotic symptoms in vivo [55]. 

## 3. Pharmacological Mechanisms of CBD; Various Molecular Pathways/Targets

CBD acts through different mechanisms, some of them unknown, and uses various molecular pathways, targets, and receptors. The most important are compiled below and summarized in Figure 2.

### 3.1. 5-HT_1A_ Receptors

5-HT_1A_ serotonin receptor also acts a molecular target for CBD. In heterologous cells expressing the 5-HT_1A_ receptor, CBD produced a dose-dependent displacement of a selective 5-HT_1A_ agonist ([3H]8-OH-DPAT) binding. Moreover, CBD at higher doses induced [35S]GTPγS binding, exhibiting an agonistic activity [59]. The selective antagonist NAN-190 was used in cAMP assay to reinforce the notion that CBD interacts with 5-HT_1A_ orthosteric binding sites in vitro. The cAMP assay assesses, in CHO cells, the percentage inhibition of forskolin-stimulated cAMP levels. The 5-HT and CBD both reduced these, while it was blocked by NAN-190, suggesting that NAN-190 competitively binds to 5-HT_1A_ receptor, competing with 5-HT and CBD for that space [60]. In vivo behavioural studies have reported that CBD acts as anxiolytic by 5-HT_1A_ dorso-lateral PAG activation, and that these actions were reversed using a 5-HT_1A_ receptor antagonist WAY100635.

### 3.2. Adenosine Receptors

CBD, along with THC, activates adenosine receptors by increasing the endogenous adenosine content, inhibiting adenosine reuptake and acting as a competitive inhibitor of the equilibrative nucleotide transporter on EOC-20 microglia cells. The A_2A_ adenosine receptor was confirmed as a CBD target, as A2A receptor antagonist, ZM 241385, and reversed the anti-inflammatory effects in a murine model of acute lung injury [41]. Moreover, in mice challenged with lipopolysaccharides (LPS), CBD significantly reduced tumor necrosis factor (TNFα), which was previously blocked by selective A2A adenosine receptor antagonist, ZM241385. Furthermore, it is also reported through the in vivo studies that CBD might also act on more than one type of adenosine receptor. CBD is shown to have antiarrhythmic effects against I/R-induced arrhythmias in rats, which was blocked by the adenosine A_1_ receptor antagonist DPCPX [49].

### 3.3. Dopamine Receptors

CBD is found to be a partial agonist of D2 dopamine receptors, as it inhibits radiolabelled domperidone binding to D2 receptors in striatal membranes of rats [61]. In a study through molecular modelling of D2 and D3 receptors in conjugation with CBD and haloperidol, it was reported that CBD binds more favorably to D3 dopamine receptors compared to D2 receptors, and acts as a partial agonist at this receptor [62]. CBD has been shown to alter the dopamine signaling in the brain. Emotionally-relevant contextual information is transmitted to the mesolimbic dopaminergic system via the ventral hippocampus (VHipp), which regulates the amount of dopamine released at the ventro-tegmental area (VTA) [63]. WIN55,212–2 (CB1 receptor agonist), when was administered systemically or via intra-VHipp injection, increases VTA dopaminergic neuronal activity and elicits dopamine efflux directly into the nucleus accumbens shell [64]. SR141716A (CB1 receptor antagonist) reverses this action. CBD exerts differential control over dopamine activity, and emotional memory processing, as it has opposing effects on molecular signaling pathways underlying schizophrenia [46,63].

### 3.4. GPR55

Apart from the extensively-discussed cannabinoid receptors CB1 and CB2, there is another cannabinoid receptor mediated through other channels, GPR55 [64]. Cannabinoid receptors agonists such as HU210, CP55940, and Δ^9^-THC specifically binds, and triggers the FLAG-tagged human GPR55 expressing heterologous cells. CBD also functions as a GPR55 antagonist. In an in vivo study, when a synthetic regio isomer of CBD named abnormal-CBD (Abn-CBD) was used, it exhibited vasodilator effects, and reduced blood pressure with no psychotomimetic effects, which proves that it can be used for parkinsonism treatment. Moreover, agonist-antagonist activity of GPR55 was confirmed as CBD blocked the anti-cataleptic property exhibited by Abn-CBD [65]. 

### 3.5. Ion Channels 

The Transient Receptor Potential Vanilloid 1 (TRPV1) receptor (known as VR1 receptor) is also a proposed molecular target for CBD. CBD increases intracellular Ca^2+^ levels, such as in the case of capsaicin, by displacing it from the TRPV1 receptor in heterogenous cells overexpressing TRPV1 receptors. It was reported that CBD acts as an agonist of this receptor [66]. CBD was also found to be modulating neuronal hyperactivity as being implicated as a molecular target in TRPV2 and TRPA1 receptors. Further studies on CBD as potential anticonvulsant and anti-epileptic activity revealed that the TRPV1 receptor was not alone, and other receptors of endocannabinoid system were also involved. This notion was further reinforced by brain activity evaluation by electroencephalograph (EEG). CBD exhibited anti-convulsant activity in a seizure-induced mice model, and further, its action was reversed by TRV1A, CB1 and CB2 receptor selective antagonists such as SB 366791, AM 251 and AM 630 [67].

### 3.6. Opioid Receptors

Vaysse et al. first postulated that CBD has modulatory effect opioid receptors, where he showed that [3H] dihydromorphine binding to MOR is decreased by Δ9-THC due to a reduction in the number of binding sites [68]. This study revealed that Δ9-THC binds in a non-competitively manner to opioid receptor as a negative allosteric modulator. Further investigations exhibited that at 30 μmol/L concentration, both THC and CBD accelerated the dissociation of [3H]-DAMGO, and [3H]-Naltrindole, from MOR and DOR in displacement binding assays performed in rat brain cortical membranes [69], and acted as negative allosteric modulators of MOR and δ opioid receptors, where increase in dissociation of THC and CBD was found to be approximately 2- and 12.47- folds respectively. Due to the close interaction between the opioid and cannabinoid systems, its endocannabinoid modulatory activity, its anxiolytic properties, and lack of psychostimulant effects, CBD can be potentially used in drug abuse and withdrawal syndrome [60]. 

### 3.7. PPARγ Receptors

Peroxisome proliferator-activated receptor gamma (PPARγ) is responsible for insulin signaling and glucose metabolism in skeletal muscles and the liver [70]. Thiazolidinediones such as rosiglitazone and pioglitazone, which are usually used in the treatment of type 2 diabetes, act as PPARγ agonists and stimulate the genes regulating insulin and fatty acid transcription, and restore the glycaemic profile in db/db mice. The db/db mice are obese, due to leptin receptor knockout, and are commonly used as models of type 2 diabetes and obesity. They have considerably higher calorie intake, hyperglycaemia, dyslipidemia, and metabolic syndrome [71]. CBD, when used in treatment of type 2 diabetes, shows lipid and glycemic parameter improvements, which shows that CBD might have agonistic activity on PPARγ. GW9662, a selective antagonist, blocked PPARγ and associated CBD effects in rat astroglial culture [72]. Moreover, CBD as PPARγ agonist may be helpful in treatment of Alzheimer’s disease. This is further supported by PPARγ agonists, e.g., pioglitazole, in improving learning and memory skills in animal models of AD [73]. Similar results were observed with rosiglitazone. Other endogenous cannabinoids such as anandamide and 2-AG also activate PPARγ and can produce anti-inflammatory responses [74].

## 4. Nanocarriers: A Potential Platform for Targeted Delivery of CBD 

Nanocarriers have been used as a potential platform for the targeted delivery of various phytocompounds including CBD. Nanodelivery systems help in improving the stability of phytocompounds, enhance their absorption, protect them from early enzymatic deprivation or metabolism within the body, and extend their circulation time, thus limiting the various adverse effects [75]. The modified nanocarriers improve the solubility and permeability, and assist in the sustained delivery of CBD to the targeted diseased sites, thus improving the bioavailability. Various studies reporting on the effective delivery of CBD through these nanocarrier systems have been discussed in this section, and briefly shown in Figure 3.

### 4.1. Lipid-Based Nanocarriers for CBD

Various lipid-based nanocarriers have been reported for the effective and site-specific delivery of CBD, including nanoliposomes, nanoemulsions, nanostructured lipid carriers (NLCs), and solid lipid nanoparticles (SLNs). The advent of CBD-loaded in lipid-based nanocarriers presented greater therapeutic effects against different diseases and disorders [9,11].

#### 4.1.1. Pro-Nanolipospheres (PNLs)

PNLs are a group of lipid-based delivery systems that are homogenous and isotropic combinations of a hydrophobic compound with a mixture of lipids, co-solvents, and surfactants. The term “pre-concentrates” is related to anhydrous liquid mixtures that after combination with an aqueous phase can form oil-in-water nanoemulsions with droplet size of 200 nm or less [76]. For the preparation of PNLs, amphiphilic co-solvent and emulsifying excipients are first completely dissolved, and then a mixture of lipids and surfactants are added and stirred to form a homogenous solution. Finally, the pre-concentrate containing CBD is added and stirred to finally form an oil-in-water nano-dispersion [10]. The hydrophobic bioactive such as CBD can be incorporated into the lipid core of obtained nanoparticles. Cherniakov et al. [10] developed advanced PNLs containing natural absorption enhancers such as curcumin, piperine, and resveratrol, which can hinder certain metabolism processes. Their results indicated that the bioavailability of incorporated CBD in advanced PNLs was enhanced up to six-fold in comparison to pure CBD [10]. Another advantage is the solubility of CBD in lipid excipients, which provides greater and constant bioavailability [76].

#### 4.1.2. Nanoliposomes

Nanoliposomes are liposomes with nano size which are suitable for the incorporation of bioactive compounds with different hydrophobicity. Nanoliposomes have a hydrophilic core surrounded by one or more phospholipid bilayers. For the preparation of nanoliposomes, dipole substances such as phospholipids are mixed with water using different mechanical (e.g., high-pressure homogenization, ultrasonication, and microfluidization), and non-mechanical (e.g., reversed-phase evaporation) methods to form nano-sized vesicles [77,78]. A nanoliposome was formed by means of a sunflower lecithin (phosphatidyl choline) base to enhance the bioavailability of CBD. Nanoliposomal CBD was around 100 nm and contained 10–20 mg/mL CBD which showed high stability at both refrigerated and room temperatures at pH 5–9 for three months [79]. In another study, liposomal CBD was formulated using hydrogenated soy phosphatidylcholine with a median particle size of 5.6 μm, containing 50 mg/g of a synthetic CBD, to improve its bioavailability for use in pain management in dogs [80]. Nanoliposomes have advantages such as the possibility of large-scale production and high targetability, as well as delivery of amphiphilic molecules but rapid release [78].

#### 4.1.3. Transferosomes

Transferosomes are a class of liposomes with high elasticity, composed of phospholipids (such as phosphatidylcholine) and different surfactants (e.g., Tween 80, sodium cholate, deoxycholate, dipotassium glycyrrhizinate, and Span 80). The surfactants are responsible for the deformability of the obtained vesicles. The formulation of transferosomes may contain a total lipid concentration of ≤10% in the final aqueous lipid suspension and also some ethanol, usually <10% [81]. Transferosomes comprising cholesterol, soya lecithin, and Polysorbate 80 were produced through thin film evaporation. The transferosomal dispersion containing CBD showed an average diameter of 102–130 nm. It was observed that the stability of CBD was improved by up to six months at room temperature after encapsulation in transferosomes [82]. 

Compared to other liposomes, transferosomes have the advantages of the liposome lipid bilayer, but addition of a surfactant that increases elasticity and deformation [81].

#### 4.1.4. Solid Lipid Nanoparticles (SLNs)

SLNs are nanoparticles containing a matrix composed of a solid lipid shell [83]. There are two methods to produce SLNs on a high scale: hot homogenization, and cold homogenization. In hot homogenization, CBD is dissolved in melted lipid and oil-in-water emulsions will be obtained after high-pressure homogenization. SLNs will form after cooling and recrystallization of the lipids in the emulsion. In cold homogenization, CBD is dissolved in melted lipid and after cooling and solidification, they are grounded to obtain lipid microparticles. Following this, the lipid microparticles are dispersed in a cold surfactant solution and then homogenized at room temperature to obtain SLNs [84]. SLNs were applied for the incorporation of curcumin and dexanabinol (a synthetic CBD-derivative) (CUR/CDB-SLNs). The mechanism associated with its antidepressant effects in corticosterone-induced cells and depression-mediated murine models was determined. In vivo results showed that CUR/CDB-SLNs significantly increased the dopamine/5-hydroxytryptamine release and decreased the corticosterone-triggered apoptosis in PC12 cells. In addition, in vivo results showed that CUR/CDB-SLNs increased levels of CB1 and CB1-mRNA, p-MEK1, and p-ERK1/2 protein expressions in the striatal and hippocampus region, thus exhibiting a neuroprotective effect, and can be used as a potential strategy for treating major depression [85]. Similar studies were performed by another group where they reported the antidepressant activity of CUR-DB SLNs in the wild-type (CBR1+/+) and CB1-knockout (CBR1–/–) mouse models of major depressive disorder [86]. CUR-DB SLNs enhanced the levels of mRNA and protein expressions, dopamine and norepinephrine release, and downregulated the expression of Rasgef1c and Egr1 genes, improving the locomotory functions of treated animals [86]. Similarly, other researchers enhanced the *trans*-membrane permeation of Δ8- THC by developing SLNs. The formulations exhibited sustained release behavior with increased drug loading studied in rabbit models. The size of SLNs assisted in the effective penetration of the encapsulated compound to the deeper ocular tissues even at a lower dose [86].

Nonetheless, SLNs allow low encapsulation loads, and in some cases, recrystallization risk exists. On the other hand, this system provides high efficiency, flexibility, and targetability [78].

#### 4.1.5. Nanostructured Lipid Carriers (NLCs)

NLCs can be produced by mixing liquid lipids with solid lipids based on the preparation method mentioned for SLNs. The obtained NLCs usually have smaller particle size compared to SLNs [87].

NLCs using stearic acid as a solid lipid and oleic acid as a liquid lipid, cetylpyridinium chloride, and Span 20 were developed for the nasal administration of CBD. Moreover, Pluronic F127 and Pluronic F68 as gelling polymers were added to the CBD-NLC dispersion to obtain a CBD-NLC-gel [88]. Monodisperse lipid nanocapsules (LNCs) as decomposable and biocompatible carriers were developed for CBD. LNCs prepared using the energetically-efficient phase inversion temperature (PIT) technique by caprylic-capric acid triglycerides, C18E15 polyethylene glycol (15) 12-hydroxystearate, soybean lecithin with 70% of phosphatidylcholine, and NaCl and after that CBD. were encapsulated into their oily core with high loading capacity [9]. Apart from decomposability and biocompatibility, NLCs offer high encapsulation loads, stability, and faster release than SLNs [78].

#### 4.1.6. Nanoemulsions

Nanoemulsions as stable colloidal delivery systems with nanosized particle structures, have a high ability in encapsulation, protection, and delivery of bioactive compounds [89]. Nanoemulsions have advantages such as the possibility of large-scale production and delivery of poorly-hydrophilic ingredients. However, this release is somehow rapid and poorly stable in gastric conditions [78]. CBD was incorporated into the oil phase of oil-in-water nanoemulsions using sonication and high-pressure homogenization. The obtained particle sizes were <200 nm and showed high stability after 30 days of production. Taskar et al. [90] reported the improved bioavailability and permeability of CBD through modified CBD-loaded nanoemulsions formed with the conjugation of amino acid and dicarboxylic acids analogs, CBD–divalinate–dihemisuccinate, and CBD–divalinate, respectively. It was noticed that the combination of both analogs significantly improved the ocular delivery of CBD, studied in a rabbit model in vivo. In addition, the CBD-dicarboxylic acids analogs exhibited better permeation of CBD through the ocular barriers. The fabrication parameters including particle size, surface charge, type of emulsifying agent, pH, and others play a crucial role in the effective delivery of CBD through nanoemulsions [90]. Chaudhari et al. [91] developed stable CBD-loaded nanoemulsions (120 nm) comprised of CBD distillates, soybean oil, and QNaturale (*quillaja saponin* containing commercial emulsifier) using high-pressure homogenization. These formulations could be efficiently used for diverse therapeutic applications [91]. 

In another study, castor oil-based nanoemulsions were developed for the effective delivery of hydrophobic drugs (fenofibrate or CBD) with improved bioavailability and stability. The method of preparation through a co-axial lamination mixer played a crucial role in regulating the stability of the nanoemulsions. Additionally, the concentration of oil and surfactant (Polysorbate 80) exhibited significant effects on the size, zeta potential, and polydispersity index of nanoemulsions [92]. CBD-loaded nanoemulsions comprised of CBD oil, vitamin E acetate, Tween-20, and ethanol were developed for improving the solubility and oral bioavailability of CBD. Results of the pharmacokinetic studies performed using a rat model showed improved oral bioavailability of CBD when administered as CBD-loaded nanoemulsions (50 mg/kg) [93]. Moreover, food-grade CBD-loaded nanoemulsions were prepared using canola oil or hemp seed oil, and medium-chain triacylglycerides. The exposure to light and the solution’s acidity were found to be two important factors responsible for maintaining the chemical stability of the CBD. As compared to bulk oil, nanoemulsions significantly improved the bioavailability of CBD [94]. 

Nanoemulsions as stable colloidal delivery systems can improve the loading, release properties, and therapeutic effects of CBD. The developed CBD-loaded nanocarriers showed improved solubility with high stability. The emulsions of medium-chain triglycerides, rapeseed oil, soybean oil, and trimyristin were prepared by high-pressure homogenization and showed high loading capacity. It was observed that the obtained emulsions showed a higher loading capacity than the suspension of SLNs based on trimyristin [95]. It was also observed that the absorption of CBD improved significantly compared with the medium-chain triglyceride (MCT) oil vehicle of CBD [96]. Several researchers investigated the effect of lipid type (medium- or long-chain triglycerides) combined with SNEDDS on the bioavailability of encapsulated CBD. It was concluded that although the type of lipid affects the solubility, bioavailability, and physicochemical properties of CBD, after incorporation in nanoemulsion formulation, its effect on the bioavailability of CBD is not easily predicted and needs more in vivo investigations [96]. 

### 4.2. Polymeric/Biopolymeric Nanocarriers for CBD

Different polymeric or biopolymeric carriers have been used for the encapsulation of CBD. Moreover, several studies have developed a complex of natural and synthetic polyester-based nanomaterials as a potential carrier for effective delivery of therapeutics with improved characteristics [97]. Polyvinylpyrrolidone (PVP), Eudragit S-100 (ES), and Eudragit E-100 (EE) polymers were also used to fabricate nano-microfibers using electrospinning to encapsulate hemp extract for biomedical applications. Duran-Lobato et al. [98] developed a novel Δ-THC-loaded PLGA NPs (TPNPs) and evaluated its anticancer potential. TPNPs demonstrated a particle size of 300 nm with high entrapment efficiency (EE) and negative zeta potential. TPNPs improved the drug release properties and exhibited sustained release behavior. Various biological assays were performed to understand the mechanism of drug targeting and the results showed that TPNPs assisted in the targeted delivery of Δ-THC into the treated cells, showing its potential in cancer therapy [99]. 

Nanomicelles of Poloxamer 407 (P407) were used for encapsulation of CBD to enhance its bioavailability and biosafety. P407 is a triblock copolymer composed of (poly)ethylene oxide (PEO) and (poly)propylene oxide (PPO) sections. It has a PEO-PPO-PEO structure and can self-assemble into a micelle structure comprising a hydrophilic shell and hydrophobic core [99]. In another study, a cannabinoid derivative was encapsulated into the nanomicelles of poly(styrene-*co*-maleic anhydride) with an average size of 152 nm. The obtained NPs had the potential for high stability and prolonged blood circulation [52]. 

Hybrid nanocarriers are composed of two or more polymers in their structure. In this regard, the polymers usually stabilize each other and can better protect the encapsulated bioactives [100]. Wang et al. [100] fabricated CBD-entrapped polymeric hybrid NPs using two polymers (zein and whey protein) by the anti-solvent precipitation technique. Zein and whey protein-based polymeric NPs significantly improved the EE of CBD with superior stability. Results of FTIR studies confirmed hydrogen bond and hydrophobic forces in complexing zein and whey protein. In addition, the zein-whey protein-CBD NPs exhibited superior antioxidant activity and improved in vitro release of CBD. The results indicated that hybrid NPs of zein and whey protein could better protect CBD against thermal degradation and UV in comparison to zein NPs [100]. 

The Millard conjugate of whey protein-maltodextrin (WP-MD) was prepared using controlled dry heating and complexed with rosmarinic acid (RA) to be used as a stabilizing agent for oil-in-water emulsions for incorporating CBD. The emulsion that was stabilized with WP-MD-RA indicated the most protection efficiency of CBD against thermal degradation and UV in comparison to the emulsion containing WP-MD, WP, and WP-RA. Sosnik et al. [101] fabricated hybrid polymeric micelles (PMs) comprised of CBD, chitosan (CS), and poly-(vinyl alcohol) (PVA). The hybrid PMs were found to be biocompatible, stable, and exhibited high penetrability of CBD within the human corneal epithelial cells [102]. Low-density lipoprotein (LDL) of yolk was complexed with carboxymethyl cellulose (CMC) as a carrier for nanoencapsulation of CBD using emulsification at neutral pH. Defatting of LDL had a significant effect on its dispersibility. It was observed that a carrier oil with low viscosity is favored and triglyceride with medium chain gives rise to better nanoencapsulation in comparison to vegetable oils such as high oleic soybean oil. 

A novel topical formulation in the form of cryogel comprising 2-hydroxyethyl cellulose and β-cyclodextrin was developed for the effective delivery of CBD. The cryogels exhibited a bi-phasic release pattern, with an initial burst release (first 3 h) followed by slower release. The cryogels showed dose-dependent cytotoxicity against human cancer cells (MJ and HUT-78), which showed its potential in the management of skin neoplastic diseases [102]. In another study, Momekova et al. [103] developed a water-soluble nanocomposite cryogel containing 2-hydroxyethyl cellulose and CBD-incorporated polymeric micelles. Indeed, CBD was first incorporated into the polymeric core-shell micelles, and then the obtained polymeric micelles were encapsulated into 2-hydroxyethyl cellulose cryogel carriers through UV-assisted cryotropic gelation [103]. 

Pickering emulsions (PEs), as surfactant-free emulsions, are a group of eco-friendly carriers for bioactive compounds. Chitosan/gum rabic (CS/GA) particles were used for the stabilization of PEs, and the degree of deacetylation (DDA) of CS was evaluated on the stability of PEs. CBD was dissolved in the oil phase of the emulsion and then added to CS/GA dispersion and homogenized. The results indicated that by increasing the DDA, the stability of PEs was increased. The skin absorption of incorporated CBD was higher than its permeation, which shows its potential use for dermal delivery of CBD [104]. In another study, a fabricated scaffold of a gelatin/nano-hydroxyapatite (G/nHAp) delivered CBD-loaded PLGA microspheres to specific sites of bone defects in a rat model [105]. Moreover, the lipid- or polymer-based nanocarrier approaches significantly improve the release behavior of CBD at the targeted site with improved loading, leading to increased bioavailability and decrease in toxicity (Figure 4).

### 4.3. Discussion of Nanocarrier Systems Available: Toxicity Concerns

Both lipid-based and polymeric nanocarriers are being investigated as delivery systems for CBD. Both types of nanocarriers have similar characteristics such as biocompatibility, biodegradability, high drug loading capacity and a long circulation time in the bloodstream, making them an attractive option for drug delivery. Both liposomes and polymeric nanocarriers have been used to deliver CBD in various studies and have been shown to increase the bioavailability of CBD, as well as the ability to target specific cells or tissues [106,107]. Polymeric nanocarriers display features that are ideal for proficient vehicles aimed at therapeutic delivery. They can be synthesized in spherical and capsules form, which are both outstanding for the delivery of drugs. Similarly, compacted lipid-based nanocarriers are beneficial for targeted delivery. Hence, employing nanocarriers for CBD delivery increases bioavailability, can decrease dosage when applied in combination with other drugs, reduces toxicity, and increases drug dissolution rate [107].

The use of CBD-loaded nanocarriers has the potential to increase the bioavailability and efficacy of CBD, as well as targeting specific cells or tissues. However, there are potential toxicity concerns associated with the use of CBD-loaded nanocarriers.

One concern is the potential for CBD-loaded nanocarriers to cause systemic toxicity. Studies have shown that CBD is generally well-tolerated and non-toxic in humans, but the use of nanocarriers may change the way CBD is metabolized and excreted. This could lead to higher levels of CBD in the body and an increased risk of toxicity [108]. This highlights the need for further studies to evaluate the safety and toxicity of CBD-loaded nanocarriers in humans.

Another concern is the potential for CBD-loaded nanocarriers to cause local toxicity. For example, when nanocarriers are delivered to the lungs or brain, they may cause inflammation and damage these target organs. Studies have shown that CBD can have anti-inflammatory effects, but the use of nanocarriers may change the way CBD interacts with cells and tissues, leading to unintended side effects. Moreover, there is also the potential for toxicity from the nanocarrier itself. Some types of nanocarriers, such as liposomes and polymeric nanoparticles, have been shown to have toxicity in animals and cells. Therefore, it is important to consider the potential toxicity of the nanocarrier itself when using CBD-loaded nanocarriers [109].

In addition, it is also important to consider the potential for interactions between CBD and other drugs when using CBD-loaded nanocarriers. CBD is known to interact with certain drugs, such as warfarin, which can lead to increased or decreased levels of these drugs in the body. Therefore, it is important to consider the potential for interactions between CBD and other drugs when using CBD-loaded nanocarriers, and to monitor patients carefully for any signs of toxicity [107,109].

In conclusion, even though CBD-loaded nanocarriers show many advantages, there are some potential concerns that should be addressed, including systemic toxicity, local toxicity, and toxicity from the nanocarrier itself. Additionally, it is important to consider the potential for interactions between CBD and other drugs when using CBD-loaded nanocarriers. More studies are needed to evaluate the safety and toxicity of CBD-loaded nanocarriers in humans.

## 5. Clinical Significance of CBD-Loaded Nanodelivery Systems (In Vitro/In Vivo Studies)

As mentioned previously, the oral bioavailability of CBD is low (~6–9%) due to its lipophilic nature, poor solubility, and high metabolization in the gastrointestinal tract [10,27]. Nanoencapsulation of CBD into different nanocarriers can address these limitations. Several researchers demonstrated the potential efficacy of encapsulated CBD in nanocarriers in preclinical and clinical treatments. 

Table 2 exhibits recent studies on the therapeutic applications of nanoencapsulated CBD. Among all the experiments, six in vitro studies and nine in vivo studies in rats, mice, and dogs were compiled.

These studies are focused on different therapeutic applications and diseases. Anti-inflammatory effects were reported in vitro and in vivo in mouse models, showing better solubility of CBD, decreased expression of inflammatory markers, and with no signs of toxicity observed [99]. Anti-cancer activity of CBD-loaded nanoparticles was evaluated on ovarian and breast cancer cell lines and mice models in combination with antitumoral drugs paclitaxel and doxorubicin, respectively. In both experiments, synergistic effects in vivo were reported, allowing reduction of the doses of the antitumoral agents, and thereby their side-effects [16,52]. Many other studies have been developed in nervous system diseases. A recent study developed in dogs researched the analgesic effects of CBD-loaded liposomes combined with analgesic supplements (robencoxib and gabapentin) and showed the potential therapeutic effect in the management of multimodal pain in dogs [80].

However, even though the latent therapeutic potential of CBD-loaded nanocarriers is significant, more studies are needed on the preclinical and clinical stage. This aspect is one of the targets for the development of safe practices.

## 6. Conclusions and Future Perspectives

CBD, an active phytochemical extracted from the *Cannabis sativa* plant, has sought the attention of many researchers due to its immense therapeutic potential. CBD has varied molecular targets, including ancillary activity over cannabinoid receptors and agonist effects with transient receptor potential (TRPV) and serotonin 1A receptor (5-HT_1A_) receptors, and a few more molecular targets are still under investigation. However, its clinical usage has been restricted due to several reasons, mainly including poor solubility, low permeability, and bioavailability. As per the reports, CBD and its derivatives have been reported to be safe, but its direct usage is still a matter of concern due to poor solubility and permeability. These limitations can be overcome with the applications of various nanocarriers including lipid-based carriers, polymerics, and others. These nanocarriers help in improving the solubility and permeability of CBD through various biological barriers, thus leading to its improved bioavailability and targetability to the diseased sites and negligible toxicity. More research about the mechanisms of action behind the effects of CBD is still needed, as well as addressing the potential for drug-drug interaction and drug-nanocarrier interaction. Clinical trials investigating the efficacy of CBD for the treatment of pain, autoimmune diseases, psychiatric disorders, substance use, and various other conditions often rely on a single acute dose, but effective doses may vary across disease states. Moreover, before recommending CBD to patients, several clinical trials should be performed to examine the effect of multiple doses administered recurrently for an extended time. Nevertheless, patients have been found to be taking self-medication of untested CBD products over-the-counter for various psychiatric and other neurological conditions. These malpractices need to be monitored at the core level. Thus, it becomes necessary to generate awareness amongst the people for CBD usage and to develop novel strategies for the effective delivery of CBD for its applications, from clinical to translational uses.

## Figures and Tables

**Figure 1 pharmaceuticals-16-00487-f001:**
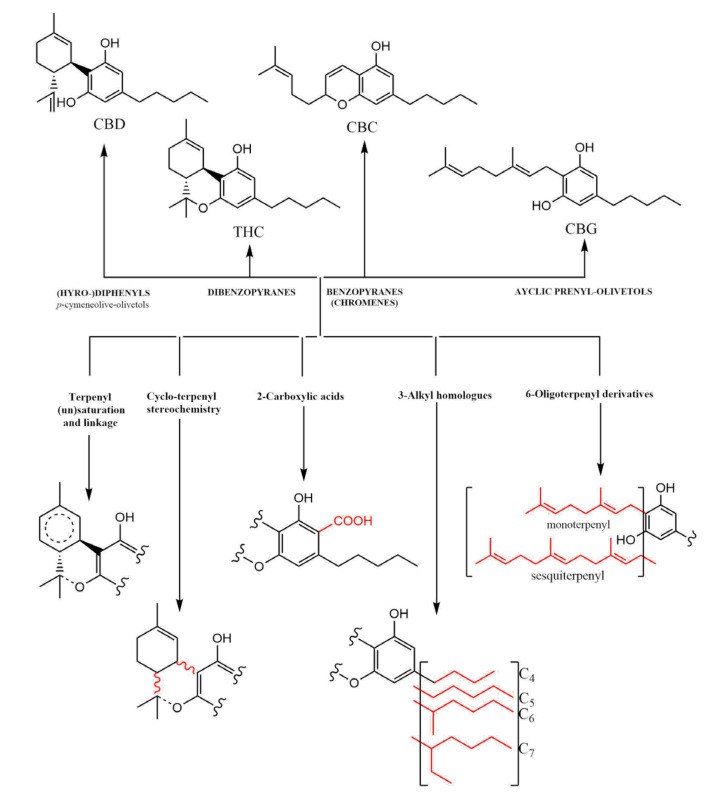
Major classification of cannabinoids (dibenzopyranes, benzopyranes, and acyclic prenyl-olivetols) based on the tetrahydro-diphenyl skeleton.

**Figure 2 pharmaceuticals-16-00487-f002:**
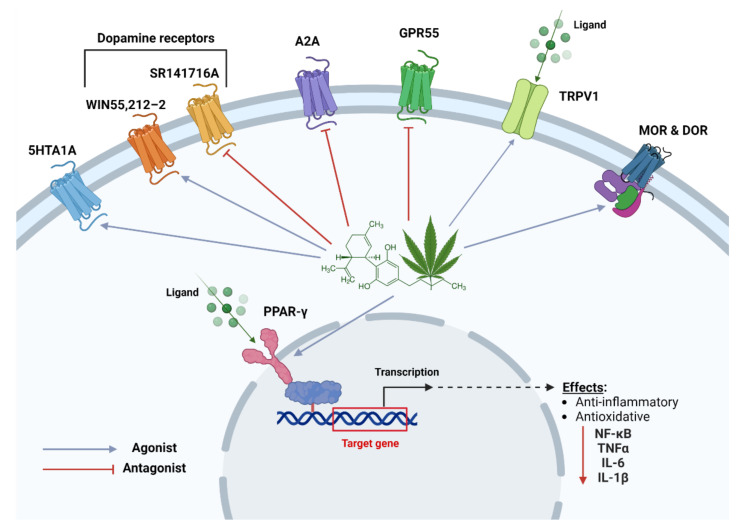
Major interactions between CBD and several membrane receptors. Blue arrows indicate agonist activity; red arrows indicate antagonist activity.

**Figure 3 pharmaceuticals-16-00487-f003:**
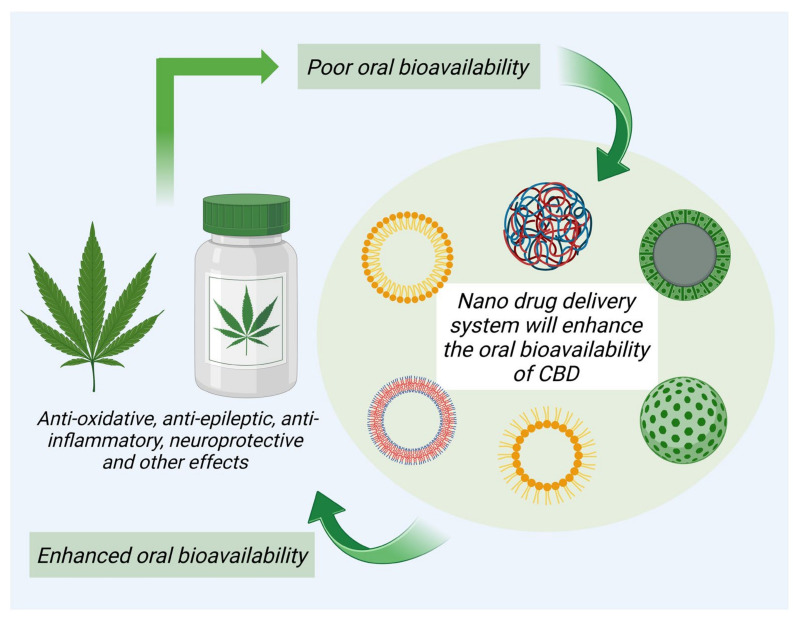
Various nanocarriers improve the targetability and therapeutic efficacy of CBD.

**Figure 4 pharmaceuticals-16-00487-f004:**
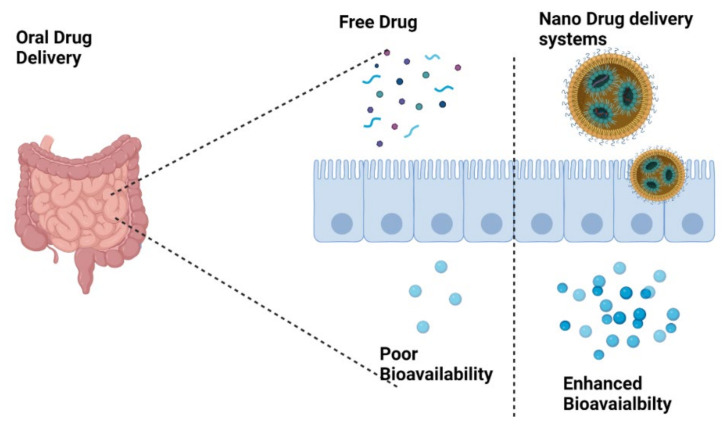
Schematic representation of release mechanisms from nanocarriers permeating via various biological barriers.

**Table 1 pharmaceuticals-16-00487-t001:** Applications and pharmacological activity of various nanoengineered CBD formulations.

Delivery System	Route of Administration	Pharmacological Activity		Application	Ref.
		In Vitro	In Vivo		
Lipid nanocarriers	Intravenous	Transport across BBB	Biodistribution in mice	Central nervous system diseases	[8]
Lipid nanocarriers	Intravenous	Cellular uptake	-	Glioblastoma	[9]
Pro-liponanospheres	Oral	-	Bioavailability in rats	-	[10]
			PK in healthy males		
Pro nano dispersions	Oral	-	PK profile in healthy males	-	[11]
Gelatin matrix pellets	Oral	-	Safety, PK profiles and relative bioavailability in healthy male	-	[12]
Gelatin matrix pellets	Oral	-	Safety, tolerability, and effectiveness in pediatric patients	-	[13]
Nanovesicles	-	-	-	-	[14]
Ethosomes	Transdermal	-	Anti-inflammatory and permeation studies in mice	Rheumatoid arthritis	[15]
Encapsulated Oil beads	Oral and Transdermal	-	Pharmacokinetic study	-	
Polymeric microparticles	-	Antitumor activity	-	Breast cancer	[16]
Poly-ε-caprolactone microparticles	-	Antitumor activity	-	Breast cancer	[17]
Poly-ε-caprolactone microparticles	Parenteral	Antitumor activity	-	Glioblastoma	[18]
α, β and γ-Cyclodextrin inclusion complexes	-	Antitumor activity	Enhanced aqueous solubility	Hepatoma/Lung adenocarcinoma	[19]
β-Cyclodextrin Inclusion complexes	Sublingual	-	Enhanced aqueous solubility and dissolution rate	-	[20]
			Enhanced dissolution and absorption		
Carbon Xerogel microspheres	-	-	Delays drug release	-	[21]
Polyoxazoline-drug conjugates	Subcutaneous	Delays drug release	Extended in vivo kinetic profile	-	[22]
Nanocrystals	-	-	Enhanced bioavailability	-	[23]

**Table 2 pharmaceuticals-16-00487-t002:** Recent studies on the therapeutic applications of encapsulated CBD.

Therapeutic Application	Carrier	In Vivo or In Vitro System	Results	Ref.
Bone healing and regeneration of critical-sized bone defects	Poly (lactic-coglycolic acid) (PLGA) microspheres	In vivo: a rat model	- A fabricated scaffold of a gelatin/nano-hydroxyapatite (G/nHAp) delivered CBD-loaded PLGA microspheres to specific sites of bone defects in a rat model - The scaffold had the ability to direct mesenchymal stem cells (MSCs) migration toward the defect site - Encapsulated CBD showed a controlled release and could noticeably enhance MSCs’ migration and improved bone healing	[105]
Glioma therapy	Lipid nanocapsules (LNCs)	In vitro: the human glioblastoma cell line U373MG	- LNCs were decorated and loaded with CBD to enhance the antitumor activity of CBD - CBD-functionalization can target any of the cannabinoid receptors that are overexpressed in glioma cell - CBD-loading along with CBD-functionalization could significantly reduce the IC50 values - Encapsulated CBD in LNCs showed prolonged release and can decrease the number of required administrations	[9]
Treatment of neuropathic pain	Nanostructured lipid carriers (NLCs)	In vivo: male Swiss mice	- NLC was used for the nasal administration of CBD to hinder liver metabolism and enhance brain bioavailability - In vivo evaluation indicated that CBD-NLC in the nasal administration had a more substantial and long-lasting antinociceptive effect than the oral or nasal administration of CBD solution in animals with neuropathic pain - The nasal administration of CBD-NLC-gel was not suitable and could not decrease mechanical allodynia	[88]
Treatment of ovarian cancer	Poly-lactic-*co*-glycolic acid (PLGA) NPs	In vitro: ovarian cancer cells (SKOV-3, OAW-42, IGROV-1) The CAM model was used for in vivo assay	- Encapsulated CBD in PLGA nanoparticles indicated a lower IC_50_ value than CBD in solution. - CBD-loaded PLGA nanoparticles are a good approach for delivering CBD intraperitoneally for ovarian cancer, and presented higher therapeutic effects than the marketed formulation - In vitro results showed improved anticancer effects of CBD against all treated ovarian cancer cells (SKOV-3, OAW-42, IGROV-1) - Combined dose of CBD+paclitaxel exhibited synergistic effects in vivo	[97]
Anti-inflammatory	Nanomicelles of Poloxamer 407 (P407)	In vitro In vivo: a mouse model	- Cell experiments indicated that anti-inflammatory markers (IL-4 and IL-10) increased, while inflammatory markers (TNF-α and IL-6) decreased - Animal experiments exhibited that inflammatory cells were repressed by CBD nanomicelles and the anti-inflammatory effect of micelles was better than that of CBD, while no obvious evidence indicated toxicity to the liver and kidney - The solubility of encapsulated CBD was enhanced	[99]
Anti-inflammatory and treatment of canine osteoarthritis pain	Liposomes	In vitro: Mouse RAW267.4 macrophage cells, primary mouse splenocytes, human monocytic THP-1 cells, and human PBMC In vivo: C57BL/6J mice	- The bioavailability of liposomal CBD was improved and thus induced potential anti-inflammatory in a mouse model in vivo - Treatment of liposomal CBD significantly reduced the level of tumor necrosis factor-α (TNF-α) in both human and mouse cell lines and can be effectively delivered through liposomal formulations	[79]
Treatment of Alzheimer’s disease	Nano chitosan	In vivo: Rat model male Wistar rats	- Induction of Alz significantly increased Aβ plaques and dead cells compared to the control group - It seems that CBD coated with nano-chitosan has good potential for reducing Aβ plaques, increasing brain CB1 and levels CB2, and improving learning and memory in Alz rats	[110]
Biocompatibility with human skin cell lines	Nanoemulsions	In vitro: human skin cell lines HaCaT keratinocytes and NHDF normal human dermal fibroblasts	- The toxicity study on human skin cell lines HaCaT keratinocytes and NHDF normal human dermal fibroblasts showed no toxicity in a wide range of concentrations - Encapsulated CBD oil in the nanoemulsion showed a positive effect on the hydration and degree of discoloration of human skin	[89]
Rectal tissue permeation	Transfersomes	In vivo: Sprague Dawley rat	- Encapsulation of CBD in the transfersomes enhanced rectal tissue permeation	[82]
Anticancer activity against triple-negative breast cancer (TNBC)	Nanomicelles	In vitro: Triple-negative breast cancer (TNBC) namely MDA-MB-231, 4 T1, and MCF-7 In vivo: Female Balb/c mice	- CBD nanoparticles were used in combination with doxorubicin (Doxo) for the treatment of TNBC and the encapsulated form of CBD improved the anticancer effect at low doses with Doxo	[52]
Analgesic treatment	Liposomes	In vivo: in dog	- In vivo analgesic activities of analgesic supplements (robencoxib and gabapentin) in a dog model (mixed breed) diagnosed with testicular neoplasia, elbow and hip osteoarthritis, and extreme cervical pain - The potential effect of subcutaneously administered liposomal CBD (5 mg/kg) was observed in the treated animals - In vivo results showed that the plasma concentration of CBD was enhanced and the liposomal CBD could be potentially used as a combinatorial therapeutic in the effective management of multimodal pain in dogs	[80]

## Data Availability

Not applicable.

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
