# Peer review of "Cannabidiol-Loaded Nanocarriers and Their Therapeutic Applications"

_pharmaceuticals, 2023, doi:10.3390/ph16040487_

Round 1
Reviewer 1 Report
The authors have briefly described the nanoformulated Cannabidiol to overcome the hydrophobicity and improve the bioavailability. The manuscript is written well, and it is clearly understandable. Definitely, the review will provide enormous data to researchers for further study and encourage more research in the same field. The present form of the manuscript should be modified following the comments given below before accepting for publication.
1. The authors have mentioned CBD's “diverse pharmacological actions” in the abstract. Authors should incorporate a sub-section stating the pharmacological actions of CBD and the mechanism, including the structural importance. How the structure affects to several disorders like neurological disorders etc. CBD Overview should address all these before going to nanoformulations. The authors have only named numerous diseases in section 2.
2. Authors can make a separate section on the Biological Effects of CBD under ‘Overview’ followed by challenges and nanoformulations to overcome.
3. The therapeutic potential section is very general. Authors are advised to enrich with more data, mechanisms etc. Authors can follow the reference https://doi.org/10.1016/j.pharmthera.2017.02.041
4. An image should be included on how the CBD interacts with membrane receptors.
Author Response
The authors appreciate the positive comments about the work and have performed all the changes suggested by the reviewer. Please check the attached document with all the responses. Please, note that in the new manuscript, the track and changes tool has been activated to follow easy the changes.

Reviewer 2 Report
This paper offers an interesting and thorough overview of the therapeutic potential of cannabidiol-loaded nanocarriers. However, I have few concerns that must be rectified.
1. The paper provides an overview of the potential therapeutic applications of cannabidiol-loaded nanocarriers, but fails to provide a comprehensive discussion of the various nanocarrier systems available.
2. The authors present a few nanocarrier systems to illustrate the use of cannabidiol-loaded nanocarriers, but the discussion of the advantages and disadvantages of each system is inadequate.
3. The paper lacks a discussion of the potential toxicity concerns associated with the use of cannabidiol-loaded nanocarriers.
4. The authors provide an overview of the various types of nanocarrier systems available, but do not provide sufficient detail on the specific properties and characteristics of each system.
5. The paper does not address the potential for drug-drug interaction and the potential for drug-nanocarrier interaction.
6. The paper does not provide sufficient detail on the preclinical and clinical studies conducted to evaluate the safety and efficacy of cannabidiol-loaded nanocarriers.
Author Response

(The authors gave the same response as above.)

Round 2
Reviewer 2 Report
Accepted in its present form